# Blood gas and lactate analysis in nesting loggerhead (*Caretta caretta*) and green (*Chelonia mydas*) sea turtles from southeastern Florida, USA

Faye E. Giebink[1], Justin R. Perrault[2], Madison Toonder[2], Sarah E. Hirsch[2], Derek M. Aoki[2], Craig A. Harms[3], Charles J. Innis[4], Nicole I. Stacy[1]*

1 Department of Large Animal Clinical Sciences, College of Veterinary Medicine, University of Florida, Gainesville, Florida, United States of America, 2 Loggerhead Marinelife Center, Juno Beach, Florida, United States of America, 3 Department of Clinical Sciences, College of Veterinary Medicine and Center for Marine Sciences and Technology, North Carolina State University, Morehead City, North Carolina, United States of America, 4 Animal Health Department and Anderson Cabot Center for Ocean Life, New England Aquarium, Central Wharf, Boston, Massachusetts, United States of America

* stacyn@ufl.edu

## Abstract

High-energy demands and transition to a catabolic state pose physiological challenges for sea turtles during the nesting season. The objectives of this study were to assess venous blood gas analytes and lactate in loggerhead (*Caretta caretta*) and green turtles (*Chelonia mydas*) nesting in southeastern Florida to establish species-specific reference intervals, examine correlations between blood analytes, and investigate differences between species. For each species, the goal was to identify associations of analytes with morphometrics, nest deposition date, and, in loggerheads, duration of emergence from the water to blood sampling. The agreement of lactate concentrations between two point-of-care analyzers (i-STAT and Nova Lactate Plus) was also compared. In total, 49 loggerheads and 30 green turtles were sampled over portions of two nesting seasons. Reference intervals were established for clinically normal nesting turtles for each species. Partial pressure of oxygen was higher in loggerheads, while partial pressure of carbon dioxide and bicarbonate were higher in green turtles. In loggerheads, lactate positively correlated with curved carapace length, while pH declined and lactate increased across the nesting season, and there were no relationships between blood analytes and time from emergence to blood collection. No morphometric or seasonal trends were observed for green turtles. There was a strong association between lactate concentrations determined by i-STAT and Nova Lactate Plus, with the i-STAT yielding higher results. The mild trend toward relative lactic acidosis across portions of the nesting season in loggerheads suggests that physiological strategies to manage periods of high-energy utilization during nesting activities vary between species. These results provide insight into the variability of blood analyte data of two species of nesting sea turtles, allow for

**Data availability statement:** All relevant data are within the manuscript.

**Funding:** Florida Department of Environmental Protection grant #21-PBL.

**Competing interests:** The authors have declared that no competing interests exists.

understanding physiological and metabolic changes during nesting, and provide relevance for clinical evaluations during health assessment studies, stranding response, and rehabilitation.

## Introduction

The Atlantic coast of Florida, USA is home to globally and regionally important aggregations of nesting loggerhead (*Caretta caretta*) and green (*Chelonia mydas*) sea turtles. Specifically, Palm Beach County, in southeastern Florida, includes several of the highest-density nesting beaches in the state [1,2]. Along with beaches in Oman and Cape Verde, northern Palm Beach County hosts the largest aggregation of nesting loggerheads in the world [1–6] and the second largest aggregation of nesting green turtles in Florida, which, in recent years, has become one of the largest in the western Atlantic Ocean [7–9]. While loggerheads are currently listed as Vulnerable and green turtles are considered of Least Concern on the International Union for the Conservation of Nature's Red List (after recent downlistings), both species remain protected under the Convention on International Trade in Endangered Species Appendix I as well as the U.S. Endangered Species Act [7,10–12]. Major threats to both loggerhead and green turtle populations include fisheries bycatch, boating interactions, illegal hunting, habitat loss, marine pollution, and disease [1,8,13–19]. Moreover, nesting sea turtles are increasingly threatened by loss and/or erosion of nesting grounds due to sea-level-rise and intensifying storm activity [20–23].

High-energy demands and transition to a catabolic state pose physiological challenges for turtles during the nesting season. Given the metabolic requirements of reproduction and the time needed to accumulate energy reserves, sea turtles typically do not nest annually [24]. Every two to three years, sexually mature females undertake breeding migrations that span hundreds to thousands of kilometers across disparate oceanic habitats [24–26] to deposit multiple clutches (5–6 avg. loggerhead [27–29]; 3–6 avg. green turtle [25,30–32]), each averaging 100–120 eggs, in a single nesting season [24–26]. Consequently, females must counterbalance energy expenditure with reproductive output. To manage these demands, both loggerheads [33,34] and green turtles [19,35] are largely considered to be capital breeders, whereby females accumulate substantial fat stores on foraging grounds and then become hyporexic or anorexic during the nesting season [36]. While feeding strategies may vary by geographical population or by individual, and despite some degree of opportunistic and/or supplemental feeding that likely occurs during internesting periods [34,37], this reduction in food intake suggests that energy reserves accumulated on foraging grounds are required not only to support individual maintenance but also vitellogenesis, migration, and nesting activities [38].

Physical activities undergone by turtles during each individual nesting event fundamentally present their own energetic demands. A single nesting event, from initial emergence to return to sea, may take anywhere from 1 to 5 hours and includes the investigation and, oftentimes, abandonment of multiple nest sites prior to successful

oviposition [39–44]. During nest digging, turtles undergo bursts of vigorous exercise followed by short, intermittent periods of rest with breathing, which subsequently trigger compensatory mechanisms to increase ventilation, heart rate, oxygen consumption, and glycogen utilization [40–42]. Physiological changes associated with these behaviors include substantially elevated metabolic rates and tendency towards metabolic acidosis, often compensated by respiratory alkalosis [40,45], hyperglycemia [43] and hyperlactatemia [40,42,43,46]. During the nesting process, green turtles experience an almost ten-fold increase in energy metabolism compared to resting levels [40]. Additionally, in green turtles, lactate increases over the course of a single nesting event and is highest in turtles sampled while returning to sea after successful completion of nesting activities [42,43], suggesting that anaerobic metabolism plays a major role in these energetically demanding behaviors.

Venous blood gases and lactate are routinely used to evaluate metabolic and respiratory status of stranded sea turtles and are useful for investigations into population health. Blood gases and lactate concentrations have been reported for successfully rehabilitated juvenile loggerheads [47] and for free-ranging immature and mature green turtles captured during health assessments in the Galápagos [48] and off the west coast of Florida [49]. Alterations in venous blood gases and/or lactate have also been used to assess physiologic stress in free-ranging [50] and captive [51] loggerheads and in free-ranging immature green turtles [52] subject to various methods of capture, forced submergence, and/or manual restraint. Given the physiological changes associated with reproduction and energetically taxing nature of nesting, acid-base status may also vary with life-stage class and therefore necessitate the establishment of baseline data specifically for nesting turtles. Venous blood gas and lactate have been published for nesting green turtles in Malaysia [46], and changes in lactate have been used to evaluate metabolic and respiratory stress during nesting for green turtles in Australia [43] and the Gulf of Oman [42]. However, blood gas and lactate reference intervals have yet to be established for any nesting loggerhead or green turtle population, nor have these analytes been examined across the course of the nesting season. These data could provide critical insight into the expenditure of energy stores and metabolic changes during reproduction and elucidate associated differences in physiologic strategies in nesting females between species.

The objectives of this study were to assess blood gas analytes and lactate in loggerheads and green sea turtles nesting in southeastern Florida to establish species-specific reference intervals, examine correlations between blood analytes, and investigate differences between species. For each species, the goal was to identify associations of analytes with morphometrics, nest deposition date, and, in loggerheads, duration of emergence from the water to blood sampling. The agreement of blood lactate values between two point-of-care (POC) analyzers, the VetScan i-STAT and the Nova Lactate Plus, was also compared.

## Materials and methods

### Ethics statement

Our study was carried out in accordance with Florida Fish and Wildlife Conservation Commission Marine Turtle Permit (MTP) #205. All handling and sampling procedures were consistent with standard veterinary protocols and best practices for sea turtles as required by the MTP [53]. Blood collection is considered minimally invasive, and all efforts were undertaken to limit the time of handling the animal and sample collection.

### Study period, site, and sample collection

All-terrain vehicles were used to conduct annual nightly patrols on adjacent Juno and Jupiter Beaches in Palm Beach County, Florida (26.943°N, −80.072°W to 26.837°N, −80.014°W) from late March through August of each year. Samples were opportunistically collected from nesting loggerheads and green turtles as they were encountered across portions of the nesting season. Both species were sampled from 4 June to 13 August 2021, with additional green turtles sampled from 10 July to 3 August 2023 to increase sample size for the generation of reference intervals. Turtles were approached during their nesting fixed action pattern once at least 50 eggs had been deposited. No turtles were tagged or handled

before eggs were laid, and sampling did not disrupt oviposition for any of the turtles in this study. For the establishment of reference intervals, only samples from the first encounter were included (i.e., one sample per turtle). Individual turtles were identified by external Inconel metal flipper tags (style 681, National Band & Tag Company, Newport, Kentucky USA) and/or internal passive integrated transponder (PIT) tags (Biomark®, Boise, Idaho USA). If either type of tag was not present, the tag was placed according to standard protocols [53].

Prior to sample collection, each turtle was subjected to visual physical examination to assess for mentation and alertness, body condition, external injuries, and/or other (including behavioral) abnormalities. Body condition was based on gross observation of neck and shoulder fat thickness [19], and external injuries were characterized as previously described for loggerheads [1,54], with mating scars classified by the presence of fresh scratches or scars on the shoulder(s) and/or cranial carapace [19]. Flexible vinyl tape measures were used to determine minimum and standard curved carapace length ($CCL_{min}$ and CCL, respectively) and curved carapace width (CCW). Additionally, time from emergence to blood draw was recorded for loggerheads. Logistical challenges prevented extended monitoring for the collection of time from emergence to blood draw in green turtles, which exhibit significantly longer nesting durations than loggerheads and can spend up to 2 hours body pitting and digging before oviposition [44].

The dorsal cervical integument was cleaned with two applications of 70% isopropyl alcohol and a blood sample of approximately 1 mL was obtained from the external jugular vein using a 5 cm 21-gauge Henke-Ject® needle (Tuttlingen, Germany) attached to a 3 mL syringe (BH Supplies, Jackson, New Jersey USA). Blood from the syringes was immediately (i.e., within 60 s of collection) placed into CG4 + cartridges and analyzed using a VetScan i-STAT POC device (Abbott Point of Care Inc., Princeton, New Jersey USA). The remaining sample volume was simultaneously analyzed using a Nova Lactate Plus POC analyzer (Nova® Biomedical, Waltham, Massachusetts USA). Blood samples were evaluated prior to analysis and excluded if plasma had greater than 1 + hemolysis, icterus, or lipemia following subsequent processing in the laboratory.

The i-STAT CG4 + cartridges directly measure pH, partial pressure of carbon dioxide ($pCO_2$), partial pressure of oxygen ($pO_2$), and lactate, then calculate base excess, bicarbonate ($HCO_3$), total carbon dioxide ($TCO_2$), and oxygen saturation ($sO_2$) using standard equations. Since pH, $pO_2$, and $pCO_2$ are temperature-dependent analytes, the i-STAT analyzes blood at 37°C. The user may then "correct" these values for body temperatures other than 37°C by entering a different temperature into the analyzer [48–50,55–57]. Since cloacal temperature of individual turtles could not be obtained due to permit restrictions, environmental temperature was used as a surrogate for body temperature [58,59]. Additionally, because the i-STAT temperature corrections use human-based algorithms, these values were also calculated manually using previously derived equations for sea turtles, as listed below [48–50,55–57]. Initial data provided by the i-STAT and temperature correction are denoted by superscripts of 'I' and 'TC,' respectively, whereas ΔT is equivalent to 37°C minus environmental temperature at the time of sample collection. While empirical decline in the slope of pH below 25°C in green turtles has prompted other studies to use different calculations for pH correction at these body temperatures [48,60,61], comparison of previously derived formulae for pH for temperatures >25°C and <25°C [61] revealed that there were no significant differences in pH (<0.5%) between equations for turtles our study. Thus, pH was calculated using the standard equation for body temperatures >25°C for all turtles regardless of environmental temperature.

$$pH^{TC} = 0.015\,(\Delta T) + pH^{I}$$

$$pO_2^{TC} = pO_2^{I}(10^{-0.0058\Delta T})$$

$$pCO_2^{TC} = pCO_2^{I}(10^{-0.019\Delta T})$$

Bicarbonate was then calculated via the Henderson-Hasselbalch equation and temperature-corrected blood gas data [48,50,62,63]. Data for base excess, $sO_2$, and $TCO_2$ were excluded from this study because i-STAT determination of base excess assumes standard human hemoglobin and plasma protein concentrations, $sO_2$ is related to species-specific distinctions in blood oxygen affinity that differ between sea turtles and mammals, between sea turtle species, and between age classes of the same sea turtle species, and $TCO_2$ is based on formulae for mammals and clinically interchangeable with $pCO_2$ [50,51].

### Statistical analyses

Statistical analyses and calculation of reference intervals were carried out using MedCalc Statistical Software (v.20.011, Ostend, Belgium). Reference intervals were established following the American Society of Veterinary Clinical Pathology reference interval guidelines [64,65]. Data were transformed using Box-Cox transformations when necessary. For each species, Pearson correlations were used for normally distributed variables (using Shapiro-Wilk tests and Q-Q plots), and Spearman correlations were applied otherwise; no outliers were identified. Ordinary least-squares regression analysis was used to determine associations of blood gas and lactate concentrations with $CCL_{min}$ and with nest deposition date for both loggerheads and green turtles and, in loggerheads, with time from emergence to blood sampling. Species comparisons for blood gas and lactate concentrations were made using independent samples or Welch's t-tests for each turtle. Lastly, agreement between whole blood lactate concentrations as determined by i-STAT and Nova Lactate Plus was assessed using Passing-Bablok regression analysis and Bland-Altman difference plots. Differences in lactate concentrations between the two methods were determined using a paired samples t-test. To establish a conversion equation between the two methods, Deming regression was performed, which accounts for measurement error in both variables, with coefficients of variation of 1.7% assigned to the i-STAT and 4.6% to the Nova Lactate Plus, yielding a λ of 2.56.

## Results

### Physical examination

In total, 49 nesting loggerheads and 30 nesting green turtles were sampled from 4 June to 13 August 2021 and 10 July to 3 August 2023. One loggerhead was sampled twice during the same nesting season: once on 9 June 2021, and a second time 42 days later on 21 July 2021. No green turtles were encountered and sampled more than once. Due to logistical challenges with i-STAT and/or Nova Lactate Plus POC devices in the field, blood samples obtained from some turtles were unable to be analyzed by either one or both instruments, resulting in varying sample numbers between analyzers.

On visual examination, all turtles were determined to be in good body condition, exhibited normal behavior and mentation, and had no observable evidence of overt traumatic injury or disease (e.g., fibropapillomatosis). Mild external injuries were noted on some individuals. Three loggerheads had nearly or completely healed carapacial injuries, likely from interaction with boat propellers, and one had minor (<5% missing) healed injuries to both hind limbs. Two loggerheads had healed amputations involving approximately 50% of the right forelimb or the right hindlimb, and another had a small, circular, partially healed wound on its neck that was approximately 5 cm deep. All green turtles showed evidence of mating scars from interactions with males. Three green turtles had carapacial scabbing or healed injuries from boating interactions, and three others had healed amputations involving approximately 20–50% of the right hind limbs. One of the turtles also had small injuries to the caudal carapace and evidence of constriction from fishing line entanglement to the right hind limb that had since healed. Given that any injuries were mild or incidental, and turtles were otherwise clinically normal, these were deemed unlikely to affect blood gas or lactate measurements.

### Reference intervals

For loggerheads and green turtles, measures of central tendency, range, and reference intervals for blood gases and lactate are reported in Table 1. Correlations between blood gases and lactate are reported in Table 2 for each species.

**Table 1. Measures of central tendency, range, and reference intervals for morphometrics, blood gases, and lactate in whole blood for nesting loggerhead (*Caretta caretta*) and green (*Chelonia mydas*) sea turtles from southeastern Florida, USA. Conventional and Standard International units are given when relevant. Parametric or robust methods for sample sizes 20 ≤ x < 120, where appropriate, were used to calculate reference intervals [64,65].**

| *Caretta caretta* | Mean±SD | Median | Range | N | RI | LRL 90% CI | URL 90% CI | Parameters |
|---|---|---|---|---|---|---|---|---|
| CCL$_{min}$ [cm] | 95.9±6.6 | 97.2 | 82.5–108.5 | 49 | – | – | – | – |
| CCL [cm] | 97.0±6.6 | 97.6 | 84.0–110.6 | 49 | – | – | – | – |
| CCW [cm] | 88.5±5.8 | 88.2 | 76.5–101.4 | 49 | – | – | – | – |
| Ambient temperature [°C] | 26.1±1.8 | 26.1 | 21.1–28.3 | 43 | – | – | – | – |
| Ambient temperature [°F] | 79.0±35.2 | 79.0 | 70.0–82.9 | 43 | – | – | – | – |
| pH$^{TC}$ | 7.513±0.060 | 7.514 | 7.381–7.662 | 43 | 7.397–7.632 | 7.372–7.423 | 7.605–7.659 | G, P, BC |
| pCO$_2$$^{TC}$ [mmHg] | 33.6±4.3 | 34.5 | 24.2–42.2 | 43 | 24.6–41.7 | 22.3–26.8 | 40.0–43.3 | G, P, BC |
| pCO$_2$$^{TC}$ [kPa] | 4.48±0.57 | 4.60 | 3.23–5.63 | 43 | 3.27–5.56 | 2.97–3.57 | 5.33–5.77 | G, P, N |
| pO$_2$$^{TC}$ [mmHg] | 58±9 | 58 | 34–82 | 43 | 40–76 | 35–44 | 72–81 | G, R, N |
| pO$_2$$^{TC}$ [kPa] | 7.7±1.2 | 7.7 | 0.1–10.9 | 43 | 5.3–10.1 | 4.7–5.9 | 9.6–10.8 | G, R, N |
| HCO$_3$$^{TC}$ [mmol/L, mEq/L] | 31.1±3.9 | 30.5 | 26.4–42.7 | 43 | 26.1–41.6 | 25.4–26.8 | 37.4–49.4 | G, P, BC |
| Lactate i-STAT [mmol/L] | 5.31±2.22 | 5.17 | 1.14–10.81 | 43 | 1.79–10.58 | 1.31–2.35 | 8.98–11.85 | G, P, BC |
| Lactate Nova [mmol/L] | 4.5±1.7 | 4.5 | 1.3–8.6 | 44 | 1.8–8.4 | 1.5–2.3 | 7.3–9.5 | G, P, BC |
| *Chelonia mydas* | Mean±SD | Median | Range | N | RI | LRL 90% CI | URL 90% CI | Parameters |
| CCL$_{min}$ [cm] | 105.3±5.4 | 105.1 | 94.4–117.9 | 30 | – | – | – | – |
| CCL [cm] | 105.9±5.3 | 106.4 | 95.0–117.9 | 30 | – | – | – | – |
| CCW [cm] | 95.9±5.0 | 95.9 | 85.2–106.1 | 30 | – | – | – | – |
| Ambient temperature [°C] | 25.6±2.4 | 26.1 | 21.1–27.8 | 21 | – | – | – | – |
| Ambient temperature [°F] | 78.1±36.3 | 79.0 | 70.0–82.0 | 21 | – | – | – | – |
| pH$^{TC}$ | 7.527±0.057 | 7.539 | 7.447–7.640 | 21 | 7.398–7.647 | 7.363–7.440 | 7.611–7.679 | G, R, BC |
| pCO$_2$$^{TC}$ [mmHg] | 40.1±4.9 | 40.6 | 28.3–49.0 | 21 | 28.0–49.2 | 22.3–33.0 | 46.9–51.2 | G, R, BC |
| pCO$_2$$^{TC}$ [kPa] | 5.35±0.65 | 5.41 | 3.77–6.53 | 21 | 3.73–6.56 | 2.97–4.40 | 6.25–6.83 | G, R, BC |
| pO$_2$$^{TC}$ [mmHg] | 28±5 | 28 | 18–40 | 21 | 18–40 | 15–21 | 36–44 | G, R, BC |
| pO$_2$$^{TC}$ [kPa] | 3.7±0.7 | 3.7 | 2.4–5.3 | 21 | 2.4–5.3 | 2.0–2.8 | 4.8–5.9 | G, R, BC |
| HCO$_3$$^{TC}$ [mmol/L, mEq/L] | 38.2±9.0 | 36.6 | 24.6–58.4 | 21 | 22.7–62.5 | 20.3–26.1 | 52.8–72.8 | G, R, BC |
| Lactate i-STAT [mmol/L] | 5.77±2.68 | 5.82 | 1.10–12.26 | 21 | 0.90–12.32 | 0.14–2.27 | 10.27–14.58 | G, R, BC |
| Lactate Nova [mmol/L] | 4.8±2.1 | 4.7 | 1.3–10.3 | 29 | 1.5–10.1 | 1.0–2.2 | 8.4–12.0 | G, R, BC |

Abbreviations: BC, Box-Cox transformation; CCL, standard curved carapace length; CCL$_{min}$, minimum curved carapace length; CCW, curved carapace width; CI, confidence interval; G, Gaussian; HCO$_3$, bicarbonate; LRL, lower reference limit; N, no transformation; P, parametric method; pCO$_2$, partial pressure of carbon dioxide; pO$_2$, partial pressure of oxygen; R, robust method; RI, reference interval; SD, standard deviation; TC, temperature-corrected; URL, upper reference limit.

Comparison of blood gas and i-STAT lactate data between nesting Florida turtles and concentrations previously reported in the literature are provided in Table 3.

## Species comparisons

Loggerheads had higher pO$_2$ (log-transform; $t(62)$ = −15.633; $P$ < 0.001; loggerhead mean ± SD = 58 ± 9 mmHg; green turtle mean ± SD = 28 ± 5 mmHg; Fig 1c), while green turtles had higher pCO$_2$ ($t(62)$ = 5.414; $P$ < 0.001; loggerhead mean ± SD = 33.6 ± 4.3 mmHg; green turtle mean ± SD = 40.1 ± 4.9 mmHg; Fig 1b) and HCO$_3$ (log-transform; $t(24.9)$ = 3.449; $P$ = 0.002; loggerhead mean ± SD = 31.1 ± 3.9 mmol/L; green turtle mean ± SD = 38.2 ± 9.0 mmol/L; Fig 1d). No

**Table 2. Results of Spearman and Pearson correlation analysis between temperature-corrected blood gases and lactate of nesting logger-head (*Caretta caretta*) and green (*Chelonia mydas*) sea turtles from southeastern Florida, USA. Correlations coefficients are provided with *P* values indicated by the asterisks.**

| *Caretta caretta* (*N*=44) | pH | $pCO_2$ | $pO_2$ | $HCO_3$ |
|---|---|---|---|---|
| $pCO_2{}^{TC}$ | −0.635*** | | | |
| $pO_2{}^{TC}$ | 0.098 | −0.165 | | |
| $HCO_3{}^{TC}$ | 0.635*** | 0.174 | −0.075 | |
| Lactate (i-STAT) | −0.666*** | 0.286 | −0.456** | −0.548*** |
| *Chelonia mydas* (*N*=21) | pH | $pCO_2$ | $pO_2$ | $HCO_3$ |
| $pCO_2{}^{TC}$ | −0.697*** | | | |
| $pO_2{}^{TC}$ | −0.273 | 0.262 | | |
| $HCO_3{}^{TC}$ | 0.918*** | −0.420 | −0.211 | |
| Lactate (i-STAT) | −0.536* | 0.190 | −0.176 | −0.616** |

*$P$<0.050; **$P$<0.010; ***$P$<0.001.

Abbreviations: $HCO_3$, bicarbonate; $pCO_2$, partial pressure of carbon dioxide; $pO_2$, partial pressure of oxygen; TC, temperature-corrected.

differences in pH (Fig 1a) or lactate concentrations as determined by i-STAT or Nova Lactate Plus were observed between species (*P*>0.050 in all cases).

## Morphometrics

A full description of morphometrics for loggerheads and green turtles is found in Table 1. Lactate as determined by i-STAT showed a weak but statistically significant association with increasing $CCL_{min}$ in loggerheads ($Lactate_{i-STAT}$=0.12*[$CCL_{min}$] − 6.47; $r^2$=0.124; *P*=0.020; *N*=43; Fig 2); however, no other correlations were observed between $CCL_{min}$ and blood gases in either species.

## Nest deposition date

In loggerheads, pH (pH=−0.001*[day of year] + 57; $r^2$=0.098; *P*=0.038; *N*=44; Fig 3a) significantly declined across portions of the nesting season, while lactate by i-STAT ($Lactate_{i-STAT}$=0.05*[day of year] − 2289; $r^2$=0.161; *P*=0.007; *N*=44; Fig 3b) and Nova Lactate Plus ($Lactate_{Nova}$=0.06*[day of year] − 2444; $r^2$=0.210; *P*=0.002; *N*=45; Fig 3c) significantly increased across portions of the nesting season but with weak associations. No seasonal trends were observed for green turtles regarding blood gases or lactate concentrations (*P*>0.050 in all cases).

## Duration of emergence to sampling

Time from emergence to blood collection in loggerheads ranged from 22 to 73 min (mean±SD=43±11 min) and was not significantly related to blood gases or lactate (all *P*>0.050).

## Lactate method comparisons

Passing-Bablok regression analysis showed a strong linear association for lactate concentrations between the two methods for both loggerheads ($r_s$=0.973; *P*<0.001; Fig 4a) and green turtles ($r_s$=0.970; *P*<0.001; Fig 4d); however, systematic (i.e., 95% confidence interval of the intercept does not contain 0) and proportional (i.e., 95% confidence interval of the slope does not contain 1) differences were observed for both species, suggesting that the two methods are not equivalent. Bland-Altman analysis indicated discordant results for both species (0.68 mmol/L for loggerheads, Fig 4b; 1.09 mmol/L for green turtles, Fig 4e), with the i-STAT yielding higher results than the Nova Lactate Plus in 36/39 (92%) loggerhead samples and 18/20 (90%) green turtle samples. A paired samples t-test indicated that i-STAT lactate concentrations were significantly higher in loggerheads ($t(38)$ = −7.233; *P*<0.001; Fig 4c) and green turtles ($t(19)$ = −5.869; *P*<0.001; Fig 4f) in

Table 3. Blood gases and lactate concentrations in loggerhead (Caretta caretta) and green (Chelonia mydas) sea turtles from this study and the literature. All studies used blood collected from the external jugular vein, except for studies 5 and 6, that used a catheter in the carotid and the brachial artery/right atrium, respectively. If plasma was used for lactate concentrations, that is indicated where relevant.

Loggerhead sea turtles (Caretta caretta)

| Study | Location | Year | Size [cm or kg] | Analysis method & sampling details | Method of capture/sampling | N | Central tendency | pH | pCO₂ [mm Hg] | pO₂ [mm Hg] | HCO₃ [mmol/L] | Lactate [mmol/L] |
|---|---|---|---|---|---|---|---|---|---|---|---|---|
| 1 | SE USA | 2001 | SCL: 50–93 | i-STAT; heparin; blood analyzed within 10 min of collection | On capture after trapped in pound net for ≤3 d | 6 | Median (10%, 90% quartiles) | 7.35 (7.27, 7.40) | 61.9 (52.2, 70.5) | 64 (60, 73) | 35.0 (32.7, 41.5) | 1.3 (1.0, 3.3) |
| | | | | | 30 min post-capture after trapped in pound net for ≤3 d | | | 7.35 (7.27, 7.38) | 42.8 (41.4, 47.4) | 67 (64, 79) | 26.5 (21.2, 29.2) | 9.5 (6.2, 13.6) |
| | | | | | On capture after <30 min trawl | 16 | | 7.16 (7.08, 7.24) | 63.0 (54.7, 85.7) | 83 (77, 93) | 22.0 (19.0, 24.7) | 15.8 (12.5, 18.1) |
| | | | | | 30 min post-capture after <30 min in trawl | | | 7.25 (7.20, 7.34) | 35.2 (32.4, 38.3) | 71 (67, 82) | 18.0 (15.2, 21.0) | 14.9 (13.1, 17.6) |
| 2 | Canary Islands | 2008–2009 | SCL: 18–50 | i-STAT; no heparin; blood analyzed "immediately" after collection | Intake (survived; rehabilitation for various causes) | 60 | Median (10%, 90% quartiles) | 7.50 (7.38, 7.59) | 30.4 (25.7, 39.9) | 71 (57, 88) | 28.7 (21.9, 34.8) | 2.0 (0.3, 7.2) |
| | | | | | Convalescent (rehabilitation for various causes) | 60 | | 7.56 (7.49, 7.63) | 33.7 (27.4, 41.0) | 59 (47, 72) | 36.5 (29.5, 44.2) | 0.3 (0.3,1.0) |
| | | | SCL: 18–61 | | Intake (died, rehabilitation for various causes) | 6 | Median (range) | 7.56 (7.05–7.68) | 30.4 (16.3–41.1) | 55 (39–68) | 33.9 (11.1–39.6) | 2.9 (0.3–18.7) |
| 3 | MA USA | 2008–2016 | SCL: 34–84 | Critical Care Express and pHOx Ultra; heparin; blood analyzed "immediately" after collection | Intake (survived, cold-stunned) | 135 | Median (range) | 7.55 (7.04–7.80) | 30 (15–57) | 69 (10–156) | 39 (27–53) | 5.1 (1.5–19.3) |
| | | | | | Convalescent (cold-stunned) | 135 | | 7.61 (7.46–7.70) | 31 (25–52) | 84 (55–118) | 38 (25–49) | 0.9 (0.2–6.3) |
| | | | SCL: 38–97 | | Intake (died, cold-stunned) | 20 | | 7.35 (7.14–7.66) | 42 (27–68) | 42 (27–68) | 38 (31–50) | 10.6 (2.8–19.9) |
| 4 | NC USA | 2017 | SCL: 19–21 | i-STAT; heparin; blood analyzed "immediately" after collection | Before manual restraint (captive yearlings) | 16 | Median (range) | 7.57 (7.44–7.70) | 50.0 (42.4–59.8) | 77 (53–88) | 32.4 (25.5–41.9)[a] | 0.3 (0.3–1.5) |
| | | | | | After 15min of manual restraint (captive yearlings) | | | 7.41 (7.25–7.50) | 59.4 (42.8–96.0) | 78 (59–100) | 27.3 (15.8–35.1)[a] | 6.5 (3.5–12.3) |
| This study | FL USA | 2021 | CCL: 83–109 | i-STAT; no heparin; blood analyzed "immediately" after collection | During oviposition | 43 | Mean±SD Median (range) | 7.51±0.06 7.51 (7.38–7.66) | 33.6±4.3 34.5 (24.2–42.2) | 58±9 58 (34–82) | 31.1±3.9 30.5 (26.4–42.7) | 5.3±2.2 5.2 (1.1–10.8) |

(Continued)

Table 3. (Continued)

**Loggerhead sea turtles (*Caretta caretta*)**

| Study | Location | Year | Size [cm or kg] | Analysis method & sampling details | Method of capture/sampling | N | Central tendency | pH | pCO$_2$ [mm Hg] | pO$_2$ [mm Hg] | HCO$_3$ [mmol/L] | Lactate [mmol/L] |
|---|---|---|---|---|---|---|---|---|---|---|---|---|
| Green sea turtles (*Chelonia mydas*) | | | | | | | | | | | | |
| 5 | Tortu-guero | 1976–1978 | Mean mass: 128 kg | Radiometer electrodes; heparin; blood ana-lyzed "promptly" after collection | At rest (captured after nesting on beach & transported to laboratory) | 5 | Mean±SEM (when reported) | 7.47 | 45.7±2.7 | 75±5 | 36.3±2.2[a] | – |
| | | | | | After 20 min of exercise (captured after nesting & transported to laboratory) | 5 | | 7.35 | 38.7±1.9 | 82±3 | 30.7±5.6[a] | – |
| | | | | | At rest after 20 min of exer-cise (captured after nesting & transported to laboratory) | | | 7.46 | 46.7±2.7 | 75±7 | 41.3±1.4[a] | – |
| 6 | Tortu-guero | NR | Mass range: 50–134 kg | Radiometer electrodes & ELISA; no information on timing from collection to analysis | At rest in supine position (collected in Costa Rica & transported to laboratory) | 9 | Mean±SD | 7.63±0.07 | 29.9±4.3 | 47±13 | 36.2±7.3[a] | – |
| 7 | Raine Island | NR | NR | ELISA; HClO$_4$ with plasma separation and freezing at −70°C | Emerging from sea | 16 | Mean±SD | – | – | – | – | 2.0±0.3 |
| | | | | | Crawling phase of nesting | | ~Mean (estimated using web plot digitizer) | – | – | – | – | ~3.0 |
| | | | | | Body pitting | | | – | – | – | – | ~5.2 |
| | | | | | Digging egg chamber | | | – | – | – | – | ~7.1 |
| | | | | | Returning to sea | | Mean±SD | – | – | – | – | 13.7±1.3 |
| 8 | Oman | 2002 | NR | Beckman Synchron CX7; K$_3$EDTA with plasma separation after three hours and storage at −80°C | Emerging from sea | 21 | Mean±SD | – | – | – | – | 9.3±0.9 |
| | | | | | Digging egg chamber | 12 | | – | – | – | – | 10.3±0.9 |
| | | | | | Returning to sea | 22 | | – | – | – | – | 15.4±1.2 |
| | | | | | Non-nesting emergence | 16 | | – | – | – | – | 8.2±0.6 |
| 9 | Galapa-gos | 2013 | CCL: 42–85 | i-STAT; heparin; blood analyzed within 10 min of collection | Wild-caught | 28 | Mean±SD (range) | 7.44±0.08 (7.26–7.57) | 49.0±9.2 (32.4–68.3) | 53±10 (36–72) | 41.1±5.6 (33.0–54.4) | 3.7±2.4 (0.8–8.7) |
| 10 | NC USA | 2014 | SCL: 33 cm | i-STAT; no heparin; blood analyzed "immedi-ately" after collection | Intake (gillnet entangled) | 1 | Value | 6.90 | 24.4 | 58 | 6.5[a] | >20.0 |
| | | | | | Recheck 4hr post-intake | | | 6.94 | 33.1 | 22 | 9.7[a] | >20.0 |

*(Continued)*

Table 3. (Continued)

**Loggerhead sea turtles (*Caretta caretta*)**

| Study | Location | Year | Size [cm or kg] | Analysis method & sampling details | Method of capture/sampling | N | Central tendency | pH | pCO₂ [mm Hg] | pO₂ [mm Hg] | HCO₃ [mmol/L] | Lactate [mmol/L] |
|---|---|---|---|---|---|---|---|---|---|---|---|---|
| 11 | Brazil | NR | CCL: 28–56 | ELISA; heparin; transport & plasma separation in laboratory | Pursuit & hand capture in water for >60 min | 34 | Mean±SD | – | – | – | – | 11.0±4.0 |
| | | | CCL: 26–43 | | Pursuit & capture by hand net for <60 min; no FP | 66 | | – | – | – | – | 5.8±4.4 |
| | | | | | Pursuit & capture by hand net for <60 min; FP present | 40 | | – | – | – | – | 8.8±3.4 |
| 12 | FL USA | 2017 | SCL: 31–47 | i-STAT; heparin; blood analyzed "immediately" after collection | Wild-caught, with & without FP | 17 | Median (range) | 7.23 (6.94–7.43) | 60.1 (43.1–77.5) | 60 (43–78) | 29.2 (22.2–36.9)[a] | 20.0 (11.1–21.0) |
| 13 | Malaysia | 2018 | CCL: 84–107 | i-STAT; heparin; blood analyzed within 10 min of collection | Post-oviposition | 19–30 | Mean±SD (range) | 7.15±0.20 (6.57–7.54)[a] | 75.2±16.2 (52.8–119.1)[a] | 97±36 (60–175)[a] | 28.1±4.6 (17.0–38.0)[a] | 13.1±3.4 (8.1–20.0) |
| This study | Florida, United States | 2021, 2023 | CCL: 95–118 | i-STAT; no heparin; blood analyzed "immediately" after collection | During oviposition | 21 | Mean±SD Median (range) | 7.53±0.06 7.54 (7.45–7.64) | 40.1±4.9 40.6 (28.3–49.0) | 28±5 28 (18–40) | 38.2±9.0 36.6 (24.6–58.4) | 5.8±2.7 5.8 (1.1–12.3) |

1: Harms et al. 2003; 2: Camacho et al. 2013; 3: Innis et al. 2019; 4: Mones et al. 2021; 5: Jackson and Prange 1979; 6: Wood et al. 1984; 7: Jessop and Hamann 2004; 8: AlKindi et al. 2009; 9: Lewbart et al. 2014; 10: Phillips et al. 2015; 11: da Fonseca et al. 2020; 12: McNalley et al. 2020; 13: Samsol et al. 2020. Abbreviations: CCL, curved carapace length; FL, Florida; FP, fibropapillomatosis; K₃EDTA, tripotassium ethylenediaminetetraacetic acid; MA, Massachusetts; NC, North Carolina; SCL, standard straight carapace length; SD, standard deviation; ELISA, enzyme-linked immunosorbent assay; NR, not reported; SCL, straight carapace length; SE, southeastern; SEM, standard error of mean.

ᵇ Data were not corrected for body temperature.

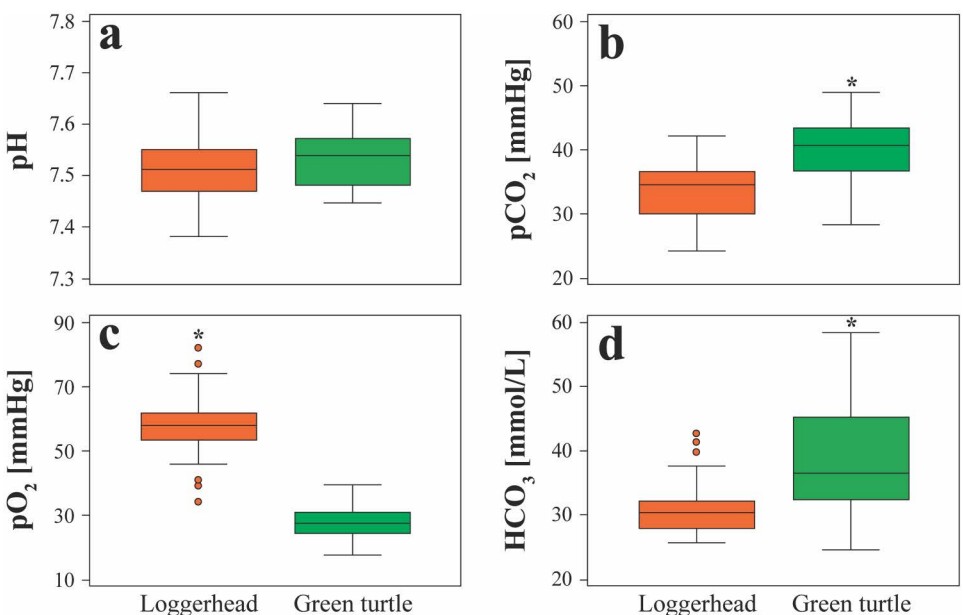

**Fig 1. Differences in (a) pH, (b) pCO₂, (c) pO₂, and (d) HCO₃ between nesting loggerhead (*Caretta caretta*) and green (*Chelonia mydas*) sea turtles from southeastern Florida, USA.** The central box represents the lower-to-upper quartile (25th–75th percentile), with the middle line representing the median. Whiskers extend from the minimum to maximum values and circles represent outside values that are larger than the upper quartile or smaller than the lower quartile plus or minus 1.5 times the interquartile range, respectively. Asterisks indicate significant differences between the two species at $P<0.050$.

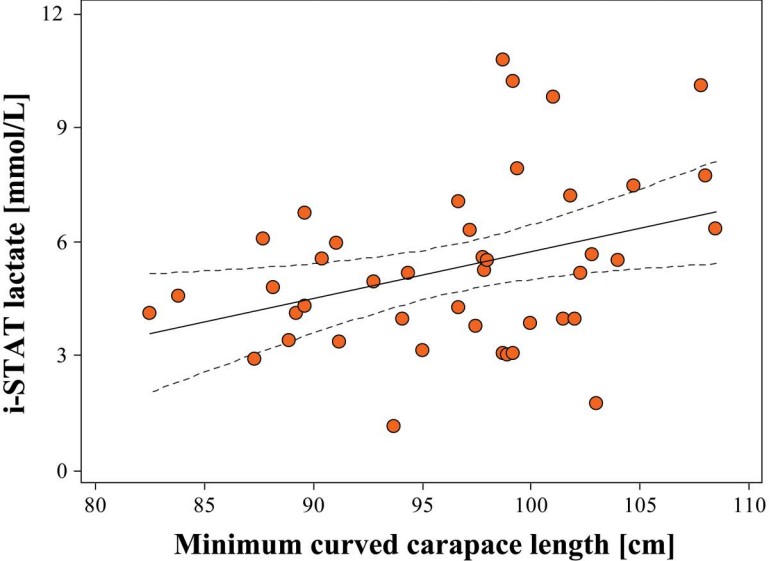

**Fig 2. Associations of minimum curved carapace length with nesting loggerhead sea turtle (*Caretta caretta*) lactate concentrations as determined by i-STAT.** The solid black line is the line-of-best-fit, while the black hatched lines represent the 95% confidence interval.

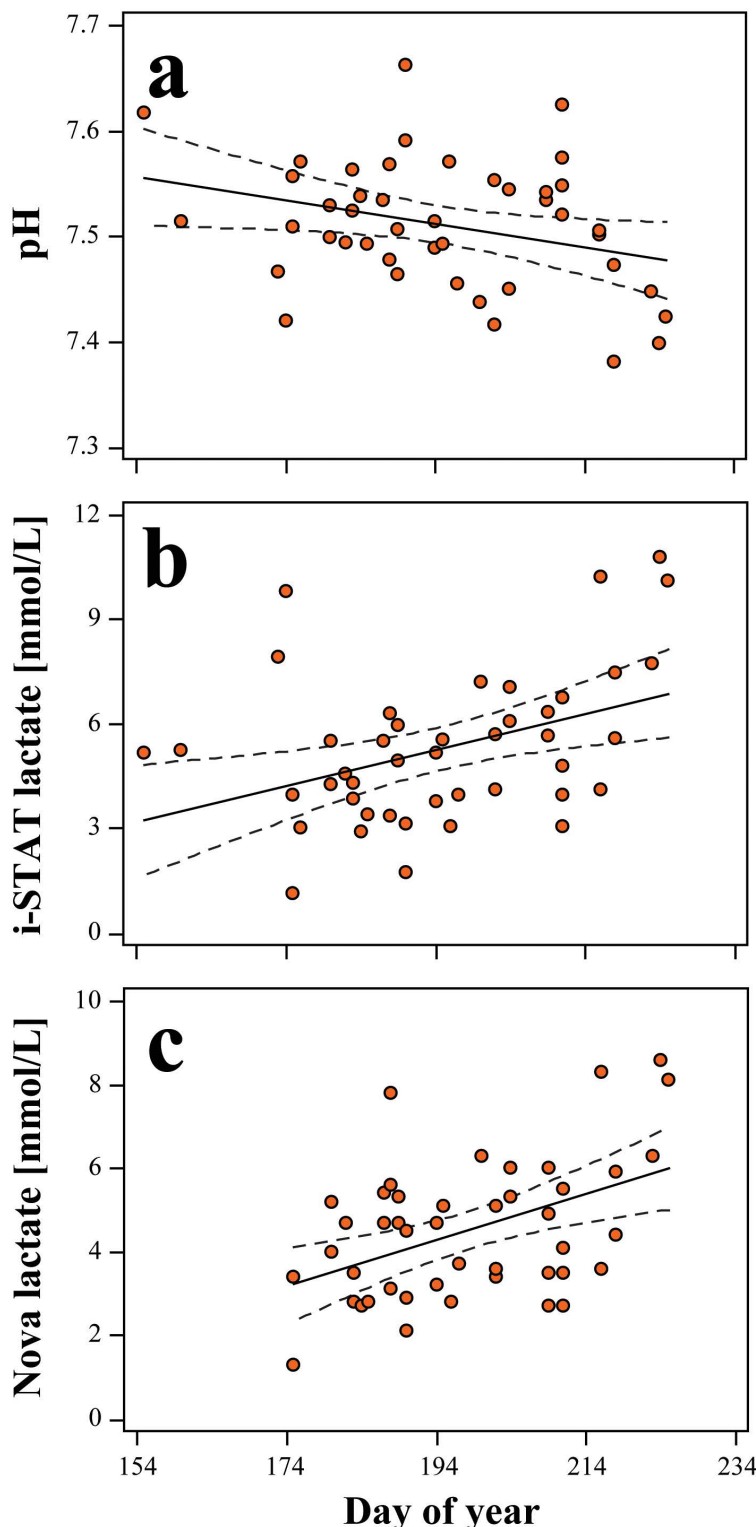

**Fig 3. Associations of nest deposition date with nesting loggerhead sea turtle (*Caretta caretta*) (a) pH and lactate concentrations as deter-mined by (b) i-STAT and (c) Nova Lactate Plus.** The solid black lines are the line-of-best-fit, while the black hatched lines represent the 95% confi-dence interval.

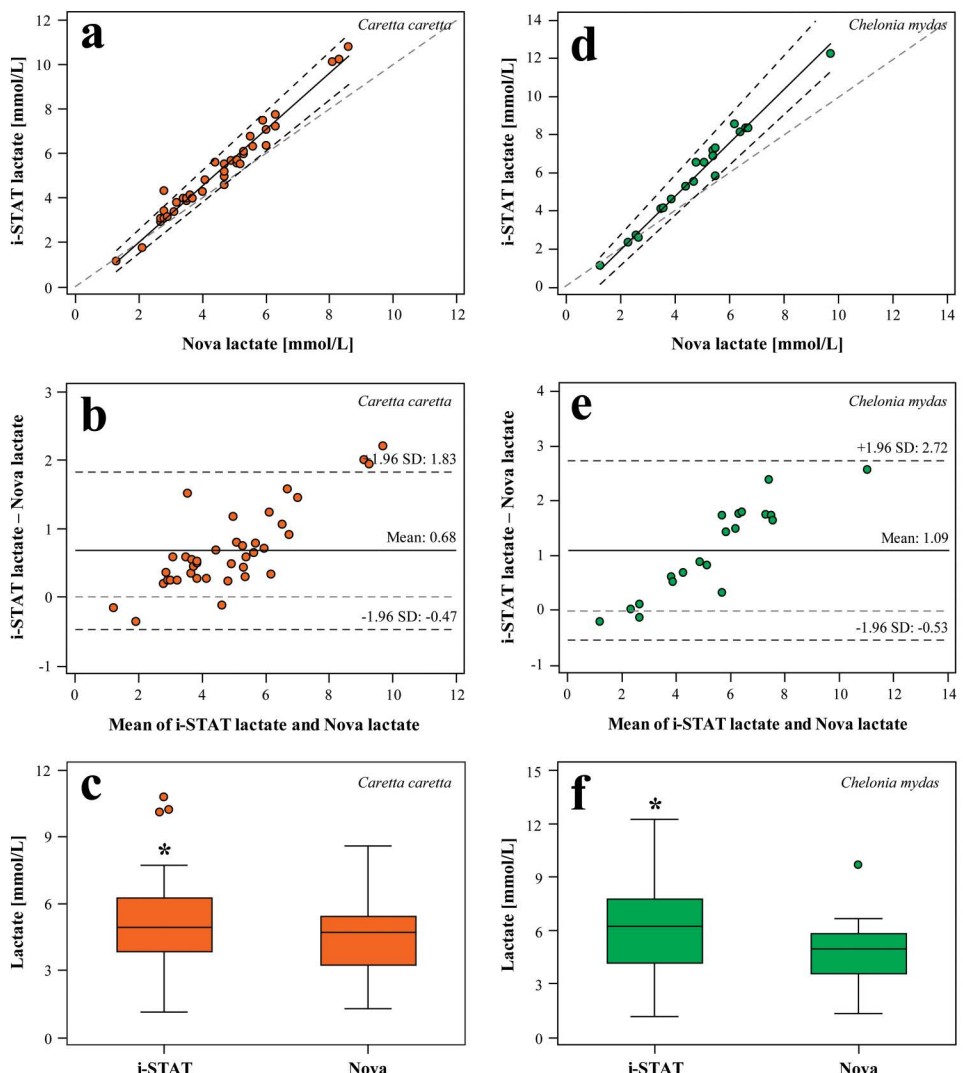

**Fig 4. Passing-Bablok regression (a, d), Bland-Altman difference plots (b, e), and paired sample t-tests (c, f) comparing lactate concentrations of whole blood in loggerhead (*Caretta caretta*) and green (*Chelonia mydas*) sea turtles, respectively, using i-STAT and Nova Lactate Plus.** For Passing-Bablok regression, the hatched gray line indicates the line of identity (y = **x**), the solid black line is the line of best fit, and the dashed black lines are the 95% confidence intervals (CIs) of the slope. In the Bland-Altman difference plots, the dashed light gray line represents the line of identity, the solid black line indicates the mean difference between the two methods, and the dark gray hatched lines are the 95% limits of agreement. An intercept close to 0, a slope close to 1, and a bias close to 0 indicate the fewest systematic and proportional differences between methodologies. Lastly, for the paired sample t-tests, the central box represents the lower-to-upper quartile (25th–75th percentile), with the middle line representing the median. Whiskers extend from the minimum to maximum values and circles represent outside values that are larger than the upper quartile plus 1.5 times the interquartile range. Asterisks above the box indicate significant differences between the two methods at *P* < 0.050.

comparison to lacta*te* by Nova Lactate Plus. Conversion equations between whole blood lactate by i-STAT and Nova Lactate Plus for loggerheads and green turtles, respectively, as generated from Deming regression analysis are as follows:

$$\text{Loggerhead}: \text{Lactate}_{i\text{–STAT}} = (1.304 \times \text{Lactate}_{\text{Nova}}) - 0.694 \ (r = 0.984)$$

$$\text{Green turtle}: \text{Lactate}_{i\text{–STAT}} = (1.413 \times \text{Lactate}_{\text{Nova}}) - 0.895 \ (r = 0.989)$$

## Discussion

This study reports blood gas and lactate data for loggerheads and green turtles nesting in southeastern Florida. It is also the first such study to examine these analytes across portions of the nesting season for both species, offering insight into the physiologic and metabolic consequences of highly energetically demanding nesting activities in sea turtles. The observed correlations between blood gases and lactate in this study suggest that nesting sea turtles experience complex interactions of metabolic and respiratory compensatory mechanisms. During nest excavation and locomotion on the beach, sea turtles undergo bursts of vigorous exercise with breath holding alternated by short, intermittent periods of rest with breathing [40]. In addition to mild relative lactic acidosis induced by muscle exertion, the potential inability to exercise and breathe simultaneously [40] may result in an increase of $pCO_2$ and thus some degree of respiratory acidosis (i.e., hypercarbia) during apneic periods. These acidotic effects are presumably counteracted by the compensatory mechanisms of respiratory alkalosis through hyperventilation and neutralization of lactic acid by bicarbonate [40,45,50].

Acid-base homeostasis is significantly impacted by metabolic rates and aerobic limits, which themselves are affected by body size, temperature, activity (e.g., exercise, diving, or muscle exertion during capture and restraint), disease, and hormonal and dietary states [49–51,66–71]. While circulating hormone concentrations implicated in heightened metabolism and energy regulation have been evaluated during different stages of the nesting process (epinephrine, norepinephrine, and corticosterone [43]; adrenaline and noradrenaline [42]; aldosterone and thyroxine [72]), and ventilation and metabolic rates have been measured during exercise in nesting green turtles [40], there has been no direct comparison to non-nesting females to quantify metabolic changes associated with reproduction and long-distance migrations from foraging to nesting areas. Apart from life-stage class, a number of distinct but interrelated factors may contribute to intra-species differences in venous blood gases and lactate, whether compared to non-nesting female turtles or to nesting turtles in other studies. These differences are likely multifactorial and may be affected by geographical location, activity type prior to capture, method of handling or restraint, and overall health and/or nutritional status. Differences in methodologies could also, in part, contribute to intra-species differences for nesting turtles, such as variations between whole blood lactate measured by i-STAT [46] versus frozen plasma lactate determined by commercial ELISA assays [42,43].

In sea turtles, both type and level of activity (e.g., swimming, diving, digging or egg-laying) just before capture may affect metabolic rate and oxygen consumption, thereby altering acid-base balance [73]. For example, leatherback sea turtles (*Dermochelys coriacea*) entangled in fishing gear had similar pH but generally higher $HCO_3$ than nesting female leatherbacks sampled during their nesting fixed action pattern, suggesting some degree of metabolic compensation during nesting [74,75]. Further, leatherbacks captured directly by boat were mildly acidotic and had the highest $pCO_2$ and lactate compared to both entangled and nesting turtles [74,75]. Because captures of foraging leatherbacks were conducted by locating animals resting at the surface of the water, it was hypothesized that turtles were actively diving and foraging immediately preceding capture, which could have induced the mild acidosis that was noted [75]. The same may be true for blood gases reported for free-ranging loggerheads and green turtles captured during health assessments or other studies wherein individuals were captured on open water. This is particularly important to consider when making comparisons to nesting counterparts. While there were minimal differences in pH between nesting green turtles and foraging Galápagos [48] and Florida green turtles [49], foraging turtles in both studies had higher $pCO_2$ than nesting turtles, indicating that they may be able to compensate exertional activities during nesting in contrast to swimming, diving, or short-term capture effects. Moreover, free-ranging loggerheads captured by trawl and pound net [50] exhibited respiratory acidosis relative to nesting loggerheads.

Independent of pre-capture activities, blood gases and lactate are possibly most affected by muscle exertion from capture, handling, and/or restraint, due to the high oxygen demand of contracting muscle tissues. For instance, lower pH and higher lactate reported for clinically normal Galápagos green turtles caught in shallow water, carried to shore, and manually restrained for blood draw [48] or for green turtles hand/dip-net captured for health assessments off the west coast of Florida [49] may have resulted from a similar stress-induced lactic acidosis as in free-ranging turtles intentionally

subjected to boat pursuit or other stressful methods of capture [50,52]. Even short periods (15 min) of manual restraint have induced significant acidosis and hyperlactatemia in loggerheads [51]. That said, turtles in this study were minimally restrained (i.e., minor adjustments of the head) for blood draw compared to routine needs for venipuncture in other settings. They were also sampled mid-oviposition during their nesting fixed action pattern, when they are minimally responsive to external stimuli [76]. In contrast, other studies of nesting turtles sampled individuals at different stages during the nesting process and/or after oviposition during their return to sea, and turtles were more heavily restrained for sampling [42,43]. Lower lactate in nesting green turtles in our study compared with studies that used significant restraint may thus be indicative of lesser handling stress [42,43]. While the magnitude of stress caused even by minimal handling during venipuncture in our study is difficult to ascertain, it is likely negligible compared to similar studies demonstrating markedly higher lactate in nesting turtles following oviposition (mean ± SD = 13.7 ± 1.3 mmol/L [43]; 15.4 ± 1.2 mmol/L [42]). This is also supported by a lesser degree of relative lactic acidosis observed in nesting turtles when compared to turtles that were purposefully subjected to capture stress to evaluate subsequent muscle exertion by the turtle [50–52].

Similar to the present study, Harms et al. (2007) sampled nesting leatherbacks on Trinidad immediately after oviposition while they remained in a state of reduced responsiveness during their nesting fixed action pattern, and blood gases and lactate were measured with the same i-STAT device used here [74]. Despite some variability in blood gases between nesting loggerheads and green turtles, mean temperature-corrected pH, $pCO_2$, $pO_2$, and $HCO_3$ of both species were not largely different from those reported for nesting leatherbacks (mean ± SD, pH 7.47 ± 0.03, $pCO_2$ 38.2 ± 3.5 mmHg, $pO_2$ 50.7 ± 14.3 mmHg, $HCO_3$ 30.0 ± 2.5 mmol/L) [74]. However, mean ± SD lactate of both nesting loggerheads and green turtles was slightly higher than lactate of nesting leatherbacks (2.5 ± 1.0 mmol/L [74]). In the present study, most venipuncture attempts were successful without redirection, suggesting that the relative degree of lactatemia observed in nesting turtles in our study was unlikely caused by venipuncture artifacts alone and may instead be related to differences in nesting behaviors between species.

Comparison of blood gases and lactate between species may support the possibility of differing physiological mechanisms to manage periods of high-energy utilization and oxygen demands during nesting. Similar lactate between nesting loggerheads and green turtles may suggest similar levels of anaerobic activity during the nesting process. However, this is unexpected given distinct species differences in nesting behavior. On average, nesting loggerheads require 63 minutes while green turtles take 146 minutes from initial emergence to return to the sea [39,44]. During body pitting and nest digging, green turtles also dig more forcefully and pause more frequently and for longer durations than loggerheads, taking up to 2 hours compared to 40 minutes, respectively, for these activities [39,44]. Green turtles may therefore require a longer overall timeframe for nesting activities, with intermittent periods of rest during longer crawls towards the dunes as compared to loggerheads with preference for nesting mid beach, to balance aerobic demands and compensate for metabolic acidosis. These considerations may also explain the higher $pCO_2$ and lower $pO_2$ in green turtles compared to loggerheads. Alternatively, similar blood lactate in both species despite behavioral differences during nesting may indicate that green turtles have a higher capacity for aerobic metabolism that compensates for more intense nesting efforts. Nesting green turtles experience an almost ten-fold increase in energy metabolism over resting levels [40], compared to a three-fold increase in nesting leatherbacks [45] and a two-to-three-fold increase in juvenile loggerheads undergoing moderate activity [67].

While the lack of data regarding nesting loggerhead metabolism precludes more direct comparisons, larger size generally correlates with greater total oxygen stores and lower metabolic rates [71]. Thus, greater average CCL (a direct correlate of body mass in sea turtles) in nesting green turtles compared to loggerheads could, in part, contribute to differences in aerobic capacity between species. If so, larger loggerheads would then also be expected to have greater total oxygen stores and capacity for aerobic metabolism than smaller loggerheads. This, however, contradicts the positive correlation between lactate and $CCL_{min}$ in nesting loggerheads, which would instead suggest a greater degree of anaerobic metabolism in larger turtles. Greater muscle mass in loggerheads likely necessitates higher oxygen demands. Comparing

these results to other studies, lactate and size (measured as body mass, straight and/or curved carapace length) positively correlated in loggerheads captured in South Carolina and Georgia, USA [50] and entering rehabilitation in Spain [47], and in Galápagos green turtles [48], but there were no correlations between size and lactate in foraging green turtles from the Gulf of Mexico [49]. Furthermore, the lack of significant correlations between time from emergence to sampling suggests that the nesting process was similarly demanding for all loggerheads in this study.

Evaluation of blood gases and lactate across a portion of the nesting season may also suggest that green turtles have more efficient adaptive mechanisms associated with energy expenditure and physical activity compared to loggerheads. In loggerheads, the inverse correlation between pH and lactate indicates a weak trend toward lactic acidosis over time. Although mild, this trend may provide additional evidence for a continuous state of catabolism in nesting females, as reduced food intake combined with high energy efforts progressively reduce somatic energy stores accumulated on foraging grounds prior to nesting migrations [33]. Decreasing glucose stores could necessitate a switch from glucose to lactate as a source for fuel. Contrary to loggerheads, stable lactate in nesting green turtles may result from similar longer crawls and nest building times typical of this species. Green turtles may utilize glucose stores at a slower rate compared to loggerheads and thus have a lesser propensity for lactate accumulation over the course of nesting season [77]. While turtles would ideally have been sampled across the entire nesting season (late April to early September and May to late September for loggerheads and green turtles, respectively), both species were still sampled across 70 days, with an additional 24 days for green turtles, during the peak nesting period, providing valuable data for temporal correlations. However, the smaller sample size in green turtles reduces statistical power and thereby limits the possibility for more conclusive interpretations regarding differing physiologic strategies to manage reproduction between both species.

Lastly, the importance of instrument validation and accuracy of blood analyzers in non-domestic animal species, including sea turtles, and under different conditions has been previously investigated [51,78–81]. Portable POC devices are particularly advantageous for field-use when studying free-ranging wildlife and provide more timely results, with less chance of artifactual errors from sample shipping and handling. The VetScan i-STAT analyzer, a portable unit that can evaluate blood gases and chemistry via disposable, single-use cartridges, is the most commonly reported POC device used to measure acid-base parameters in sea turtles (Table 2). A single lactate and blood gas cartridge is moderately costly; moreover, the device has a narrow operational temperature range (16–27°C) and is reportedly sensitive to environmental conditions such as temperature and humidity [51,78]. The Nova Lactate Plus analyzer is a lower-cost alternative that utilizes a simple test strip and operates over a wider temperature range of 5–46°C. Mones et al. (2021) evaluated the Nova Lactate Plus for use in sea turtles and concluded that despite excellent correlation between i-STAT and Nova Lactate Plus analyzers, the two methods cannot be used interchangeably [51]. The present study produced similar results using a larger sample size across two sea turtle species. We report a very strong and significant relationship between whole blood lactate concentrations determined by i-STAT and Nova Lactate Plus POC devices for both loggerhead and green turtles, with i-STAT yielding higher results than Nova Lactate Plus [51]. Disagreement between devices increased with higher lactate concentrations, thereby confirming that methodology-specific reference intervals are necessary. We thus provide analyzer-specific intervals for both loggerheads and green turtles to allow for the utility of either POC device for the monitoring of trends in clinical and research settings for each species. We also provide conversion equations to estimate i-STAT concentrations from Nova Lactate Plus measurements; however, these should be interpreted with caution, as the analysis revealed both proportional and systematic bias between devices.

## Conclusions

This study provides insight into the acid-base status of loggerheads and green turtles during a single nesting event and across portions of the nesting season, during which sea turtles expend critical energy stores and undergo phases of high physical demands while having no or reduced food intake. Both loggerheads and green turtles experience complex interactions of metabolic and respiratory compensatory mechanisms, and a trend toward mild lactic acidosis across portions

of the nesting season in loggerheads may indicate that physiological strategies to manage periods of high-energy utilization during nesting vary between species. The establishment of reference intervals for blood gases and lactate in nesting sea turtles are essential tools for understanding physiologic mechanisms during nesting and for expanding on diagnosis, prognosis, and monitoring of these species in both clinical and research settings, including population health assessment studies and stranding responses. This study underscores the importance not only of developing both species- and cohort-specific reference intervals, but also of considering methodology differences when comparing studies. The data herein give insight into the variability of blood analyte data that might be expected within a given species and life-stage class, help inform the identification and quantification of various stressors or disease, and may ultimately be utilized for decisions in sea turtle management and conservation.

## Acknowledgments

The authors wish to thank team members of the Loggerhead Marinelife Center for their technical assistance. We also thank Meghan Koperski for assistance with permitting.

## Author contributions

**Conceptualization:** Faye E. Giebink, Justin R. Perrault, Nicole I. Stacy.

**Data curation:** Faye E. Giebink, Justin R. Perrault, Madison Toonder, Sarah E. Hirsch, Derek M. Aoki, Nicole I. Stacy.

**Formal analysis:** Justin R. Perrault.

**Funding acquisition:** Justin R. Perrault.

**Investigation:** Faye E. Giebink, Justin R. Perrault, Sarah E. Hirsch, Derek M. Aoki, Craig A. Harms, Charles J. Innis, Nicole I. Stacy.

**Methodology:** Faye E. Giebink, Justin R. Perrault, Madison Toonder, Sarah E. Hirsch, Derek M. Aoki, Craig A. Harms, Charles J. Innis, Nicole I. Stacy.

**Project administration:** Justin R. Perrault, Nicole I. Stacy.

**Resources:** Justin R. Perrault, Nicole I. Stacy.

**Software:** Justin R. Perrault.

**Supervision:** Justin R. Perrault, Nicole I. Stacy.

**Validation:** Faye E. Giebink, Justin R. Perrault, Derek M. Aoki, Craig A. Harms, Charles J. Innis, Nicole I. Stacy.

**Visualization:** Justin R. Perrault.

**Writing – original draft:** Faye E. Giebink, Justin R. Perrault, Nicole I. Stacy.

**Writing – review & editing:** Faye E. Giebink, Justin R. Perrault, Madison Toonder, Sarah E. Hirsch, Derek M. Aoki, Craig A. Harms, Charles J. Innis, Nicole I. Stacy.

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
