## [Decision Letter · Decision Letter 0]

25 Nov 2025

Dear Dr. Stacy,

Thank you for submitting your manuscript to PLOS ONE. After careful consideration, we feel that it has merit but does not fully meet PLOS ONE’s publication criteria as it currently stands. Therefore, we invite you to submit a revised version of the manuscript that addresses the points raised during the review process.

We look forward to receiving your revised manuscript.

Kind regards,

Vitor Hugo Rodrigues Paiva, Ph.D.

Academic Editor

PLOS ONE

**Journal Requirements:**

1. When submitting your revision, we need you to address these additional requirements. Please ensure that your manuscript meets PLOS ONE's style requirements, including those for file naming. The PLOS ONE style templates can be found at https://journals.plos.org/plosone/s/file?id=wjVg/PLOSOne_formatting_sample_main_body.pdf and https://journals.plos.org/plosone/s/file?id=ba62/PLOSOne_formatting_sample_title_authors_affiliations.pdf 2. To comply with PLOS One submissions requirements, in your Methods section, please provide additional information regarding the experiments involving animals and ensure you have included details on (a) methods of sacrifice, (b) methods of anesthesia and/or analgesia, and (c) efforts to alleviate suffering. 3. Thank you for stating the following financial disclosure: Florida Department of Environmental Protection grant #21-PBL   Please state what role the funders took in the study.  If the funders had no role, please state: "The funders had no role in study design, data collection and analysis, decision to publish, or preparation of the manuscript." If this statement is not correct you must amend it as needed. Please include this amended Role of Funder statement in your cover letter; we will change the online submission form on your behalf. 4. If the reviewer comments include a recommendation to cite specific previously published works, please review and evaluate these publications to determine whether they are relevant and should be cited. There is no requirement to cite these works unless the editor has indicated otherwise. 

Reviewers' comments:

**Comments to the Author**

1. Is the manuscript technically sound, and do the data support the conclusions?

Reviewer #1: Yes

Reviewer #2: Partly

2. Has the statistical analysis been performed appropriately and rigorously?

Reviewer #1: Yes

Reviewer #2: Yes

3. Have the authors made all data underlying the findings in their manuscript fully available?

Reviewer #1: Yes

Reviewer #2: No

4. Is the manuscript presented in an intelligible fashion and written in standard English?

Reviewer #1: Yes

Reviewer #2: Yes

**Reviewer #1:** This is a very well written article with valuable information regarding the effects of life stage and species on blood gas and lactate measurements in sea turtles. There are a few small areas that I believe could be clarified.

1. You state that the animals were less likely to have stress as a factor affecting blood sampling for these animals as they were in the nesting fixed action pattern. Did you measure the length of time the animals were handled prior to blood draw? Previous data shows that even 15 minutes of handling can significantly alter blood gas and lactate levels so time that your animals were handled seems relevant.

2. Fig 2 - pO2 was significantly different in loggerheads and green turtles - I don't see that this was discussed anywhere. Venous pO2 does not correlate with arterial pO2 and is generally thought to be not clinically relevant. Therefore there is not much to discuss but if it is in a figure it should be discussed at least briefly.

3. p22 end of paragraph correlations between blood gas and lactate (for some reason line numbering does not extend past p11) - you state the acidotic effects are counteracted by neutralization of biocarbonate through lactic acid. Is this true? Generally metabolic acid base compensation is regulated by bicarbonate metabolism through the kidneys. I am not aware of lactic acid being part of this - is this a species difference or is it not known?

**Reviewer #2:**  Summary of the Manuscript

This manuscript presents blood gas and lactate measurements from nesting loggerhead and green turtles using two point-of-care devices. The authors analyse the data separately by species, comparing loggerheads and green turtles and then examining body-size relationships, analyte correlations, agreement between devices, and temporal patterns within each species (with significant size relationships and seasonal patterns observed in loggerheads). The resulting species-specific reference intervals are clinically and ecologically useful and help fill an important gap, particularly for field teams working with nesting turtles.

See attachment for the remainder of the Review

**Do you want your identity to be public for this peer review?** For information about this choice, including consent withdrawal, please see our Privacy Policy

Reviewer #1: **Yes:** Valerie Johnson

Reviewer #2: No

---

## [Author Response · Author response to Decision Letter 1]

12 Jan 2026

January 12, 2026

Dear Editor and Reviewers,

Please find enclosed our revised manuscript entitled “Blood gas and lactate analysis in nesting loggerhead (Caretta caretta) and green (Chelonia mydas) sea turtles from southeastern Florida, USA” for consideration of publication as a research article in Plos ONE. We collectively thank you for your constructive, insightful and well-reasoned feedback of our manuscript.

We have provided two revised versions of the manuscript, one showing track changes and one with these changes accepted. Additionally, we have replied individually to each point raised by the reviewers; please see responses below. Line numbers refer to the track changes version when “all markup” is selected for review (please note: the line numbering does appear non-continuous unless “simple” or “no markup” is selected due to a limitation with Word itself, so the line numbers are unfortunately inconsistent).

In sum, multiple changes have been made throughout the manuscript to address the reviewers’ feedback, and the abstract has also been revised accordingly. Of particular importance, we have clarified some of the methodology and results, including sample collection across portions of the nesting season, turtle handling, physical examination, sample quality control, temperature-correction of blood gases, and statistical analyses. Moreover, we have reorganized and consolidated the discussion section, including removal of subheaders, to better integrate and hopefully present a clearer analysis of the data to the reader. As suggested by one reviewer, we have also clarified terminology and edited wording such that the discussion of species differences in regard to metabolic capacity are presented in a more presumptive manner.

Thank you again for your time, feedback, and consideration of this manuscript. Thank you for your continued considering our manuscript for publication. We look forward to hearing from you soon.

Sincerely,

Faye E. Giebink, Justin R. Perrault, Madison Toonder, Sarah E. Hirsch, Derek M. Aoki, Craig A. Harms, Charles J. Innis, and Nicole I. Stacy

Reviewer #1:

This is a very well written article with valuable information regarding the effects of life stage and species on blood gas and lactate measurements in sea turtles. There are a few small areas that I believe could be clarified.

Response: Thank you for your positive feedback and review of our manuscript.

1. You state that the animals were less likely to have stress as a factor affecting blood sampling for these animals as they were in the nesting fixed action pattern. Did you measure the length of time the animals were handled prior to blood draw? Previous data shows that even 15 minutes of handling can significantly alter blood gas and lactate levels so time that your animals were handled seems relevant.

Response: Thank you for these considerations. We did not measure the exact length of time the animals were handled prior to blood draw. However, the entire process of the physical exam, morphometric measurements, and venipuncture were generally under 15 minutes. The cited study (Mones et al. 2021) showing significant acidosis and hyperlactatemia in turtles within 15 minutes was performed on “normal” “awake” turtles (i.e., not turtles that are in a nesting fixed action pattern, which is a sort-of “trance” in which nesting turtles are minimally responsive to external stimuli during oviposition, as was done for our study). Given this difference in the physiologic state of turtles between studies, it makes more exact comparison regarding the length of time that turtles were handled difficult.

2. Fig 1 - pO2 was significantly different in loggerheads and green turtles - I don't see that this was discussed anywhere. Venous pO2 does not correlate with arterial pO2 and is generally thought to be not clinically relevant. Therefore, there is not much to discuss but if it is in a figure, it should be discussed at least briefly.

Response: Thank you for making this observation! This has been added to the discussion (p. 22; lines 1053-1117): “Green turtles may therefore require a longer overall timeframe for nesting activities, with intermittent periods of rest during longer crawls towards the dunes as compared to loggerheads with preference for nesting mid beach, to balance aerobic demands and compensate for metabolic acidosis. These considerations may also explain the higher pCO2 and lower pO2 in green turtles compared to loggerheads.”

3. P22 end of paragraph correlations between blood gas and lactate (for some reason line numbering does not extend past p11) - you state the acidotic effects are counteracted by neutralization of bicarbonate through lactic acid. Is this true? Generally metabolic acid base compensation is regulated by bicarbonate metabolism through the kidneys. I am not aware of lactic acid being part of this - is this a species difference or is it not known?

Response: Yes, this is correct. The neutralization of lactic acid by bicarbonate is a phenomenon that occurs during anaerobic metabolism in both humans and other animals. There are multiple mechanisms by which the body achieves acid-base homeostasis: (1) neutralization via buffer systems (both intracellular and extracellular), (2) exhalation by the respiratory system, and (3) clearance by the renal system. (Shaw and Gregory 2022 succinctly reviews the physiology of acid-base balance: https://www.sciencedirect.com/science/article/pii/S2058534922000816). During strenuous exercise, for example, lactic acid accumulates in the skeletal muscle and rapidly dissociates into lactate and hydrogen ions, which decrease pH; bicarbonate ions in the blood then bind hydrogen to form carbonic acid and pH is increased. Since the wording of this sentence might have been confusing, we have changed the wording to (p. 19; lines 866-868): “These acidotic effects are presumably counteracted by the compensatory mechanisms of respiratory alkalosis through hyperventilation and neutralization of lactic acid by bicarbonate.” We have also corrected the issue regarding line numbering; however, please note: the line numbering does appear non-continuous unless “simple” or “no markup” is selected due to a limitation with Word itself, so the line numbers are unfortunately inconsistent.

Reviewer #2:

Summary of the Manuscript:

This manuscript presents blood gas and lactate measurements from nesting loggerhead and green turtles using two point-of-care devices. The authors analyze the data separately by species, comparing loggerheads and green turtles and then examining body-size relationships, analyte correlations, agreement between devices, and temporal patterns within each species (with significant size relationships and seasonal patterns observed in loggerheads). The resulting species-specific reference intervals are clinically and ecologically useful and help fill an important gap, particularly for field teams working with nesting turtles.

Overall Recommendation:

Major Revision. The study is valuable, but key parts of the manuscript need clearer explanation, more cautious interpretation, and some restructuring. A number of the physiological claims go a bit beyond what the data can actually support, and several methodological pieces need better transparency. The Discussion is also quite long and at times feels like a literature review that does not always relate back to the data and is sometimes quite speculative.

Response: Thank you for your constructive feedback! As per your detailed comments below, we have revised the entire manuscript, particularly to clarify methodology, consolidate the discussion section, and overall changed some wording to hopefully make things more understandable to the reader. Given the concerns regarding the discussion, we have also adjusted the wording in several sentences to lessen the “definitiveness” of the conclusions and ensure the reader understands that discussion of species differences is presumptive given the current data and warrants further study with larger sample sizes.

General Comments:

1. Several parts of the manuscript would benefit from clearer explanation and slightly more cautious interpretation of the results. First, the sample sizes need to be made more transparent. While 49 loggerheads and 30 green turtles were initially sampled, the final analyte-specific numbers are noticeably lower (e.g., loggerheads n = 43–44; greens n = 21). It would help readers if the authors clarified that only turtles with usable blood values were included for each analyte, and briefly explained why the numbers vary across variables (e.g., incomplete cartridges, missing analytes, sample-quality issues). This is particularly important because the reference intervals and species comparisons are based entirely on these final datasets.

Response: Thank you for this suggestion. This has been clarified in results (p. 10; lines 453-456): “Due to logistical challenges with i-STAT and/or Nova Lactate Plus POC devices in the field, blood samples obtained from some turtles were unable to be analyzed by either one or both instruments, resulting in varying sample sizes between analyzers.”

2. A second major point relates to the repeated use of phrases such as “across the nesting season.” The current sampling structure doesn’t quite support this wording. Loggerheads were sampled across the full June–August 2021 period, but green turtles were sampled across two different years, with the 2023 individuals coming only from mid- to late season. As a result, the temporal coverage isn’t parallel between species, which could affect the ability to detect any seasonal patterns in green turtles. It’s also unclear whether loggerheads and greens have the same nesting phenology at this site, or how the timing of sampling aligns with each species’ natural nesting peak. Only the loggerhead temporal analysis is shown (Fig. 3); the green turtle regressions are not presented, even though they are described as non-significant. Given this uneven temporal coverage and the mixing of years, the statements about physiological patterns “across the nesting season,” and the idea that species differ in their seasonal physiological strategies, feel a bit stronger than the data can support. A more cautious phrasing here, along with a brief acknowledgement of these sampling limitations and potential inter-annual differences, would help readers interpret the findings more accurately.

Response: Thank you for your suggestion. We have changed the wording “across portions of the nesting season” to qualify this concern. We have also changed some other wording to indicate that the relationship is weak and also added discussion of this possible limitation to the end of the discussions section (p. 24; lines 1345-1351): “While turtles would ideally have been sampled across the entire nesting season (late April to early September and May to late September for loggerheads and green turtles, respectively), both species were still sampled across 70 days, with an additional 24 days for green turtles, during the peak nesting period, which provides valuable data for temporal correlations. However, the smaller sample size in green turtles reduces statistical power and thereby limits the possibility for more conclusive interpretations regarding differing strategies to manage reproduction between both species.”

3. There are also several behavioural and reproductive explanations—such as differences in clutch size, number of eggs laid, digging effort, duration on the beach, or pauses during nest construction—that are used to interpret physiological differences between species. However, none of these behavioural variables were measured in this study. It also isn’t shown whether clutch size actually differs between species in this dataset. More importantly, the physiological patterns don’t line up neatly with these proposed explanations (for example, loggerheads had higher pO₂, while greens had higher pCO₂ and bicarbonate). Since these behaviours weren’t quantified and the proposed mechanisms aren’t directly supported by the measured variables, I think these interpretations need to be presented as speculative rather than definitive.

Response: Thank you for these considerations, but we respectfully disagree with this feedback since the behavioral and reproductive aspects are well described in the literature and consistent by species. Therefore, measuring the behavioral variables and differences between species would have been beyond the scope of this study and not needed. The same applies for clutch sizes and other end points; nevertheless, we do not make any interpretations regarding any possible effects of clutch size or number of eggs laid on species differences in blood gases or lactate. The behavioral interpretations are speculative as we wish to present the reader with multiple possible interpretative explanations for the differences noted between species. As suggested by the reviewer, we have therefore edited some of the wording to make these claims less definitive (e.g., using words such as “may” – p. 22, lines 1051, 1059; or “could” – p. 24, line 1341, etc).

4. Finally, parts of the Discussion feel quite long and occasionally drift into general literature review or physiological speculation that doesn’t always tie back to the actual findings. Tightening up some of these sections, keeping the focus on what was actually measured, and making sure interpretations remain grounded in the presented data would strengthen the manuscript considerably. There are also a few places where terminology—especially “nesting season” versus “season”—isn’t used consistently, and some key terms aren’t defined. A quick cleanup of terminology would improve clarity, particularly where temporal interpretations are involved.

Response: Thank you for your feedback. We have addressed this throughout the discussion. As suggested, we have also changed the wording “across portions of the nesting season” to qualify this concern.

5. In addition, the statistical methods section would benefit from a bit more clarity around how normality was assessed, how choices were made between parametric and non-parametric tests, and how outliers and reference-interval methods were handled. Some of this information appears later in the Results, but including a brief explanation in the Methods would help with overall transparency (see Methods and Results for more detail).

Response: Thank you for your feedback. The statistical methods section has been clarified as suggested here and, in more detail, below – for example (p. 9, lines 375-382): “For each species, Pearson correlations were used for normally distributed variables (using Shapiro-Wilk tests and Q-Q plots), and Spearman correlations were applied otherwise; no outliers were identified. Ordinary least-squares regression analysis was used to determine associations of blood gas and lactate concentrations with CCLmin and with nest deposition date for both loggerheads and green turtles and, in loggerheads, with time from emergence to blood sampling. Species comparisons for blood gas and lactate concentrations were made using independent samples or Welch’s t-tests for each turtle.”

6. In addition, a light editorial pass would be helpful. Some sentences, especially in the Introduction, Methods and Discussion, are quite long and rely heavily on parentheses, which can interrupt the flow. It may also help to standardise terminology and abbreviations (e.g., “point-of-care,” “POC,” “Point-of-Care”; “i-STAT,” “i-STAT 1,” “i-STAT one”) and to use one consistent form throughout the text, tables and figure captions.

Response: Edited as suggested. These terms have been standardized throughout the manuscript.

Specific Comments:

ABSTRACT

L26–27: Sample size clarification as above — it should be clear that only turtles with usable analytes were included, and the final n varies.

Response: Edited as suggested. This has been clarified in results (p. 10, lines 453-456): “Due to logistical challenges with i-STAT and/or Nova Lactate Plus POC devices in the field, blood samples obtained from some turtles were unable to be analyzed by either one or both instruments, resulting in varying sample sizes between analyzers” and the abstract has been reworded

---

## [Decision Letter · Decision Letter 1]

11 Feb 2026

Blood gas and lactate analysis in nesting loggerhead (Caretta caretta) and green (Chelonia mydas) sea turtles from southeastern Florida, USA

PONE-D-25-54442R1

Dear Dr. Stacy,

We’re pleased to inform you that your manuscript has been judged scientifically suitable for publication and will be formally accepted for publication once it meets all outstanding technical requirements.

Kind regards,

Vitor Hugo Rodrigues Paiva, Ph.D.

Academic Editor

PLOS One

Additional Editor Comments (optional):

Reviewers' comments:

Reviewer's Responses to Questions

**Comments to the Author**

Reviewer #1: All comments have been addressed

Reviewer #2: All comments have been addressed

2. Is the manuscript technically sound, and do the data support the conclusions?

Reviewer #1: Yes

Reviewer #2: Yes

3. Has the statistical analysis been performed appropriately and rigorously?

Reviewer #1: Yes

Reviewer #2: Yes

4. Have the authors made all data underlying the findings in their manuscript fully available?

Reviewer #1: Yes

Reviewer #2: No

5. Is the manuscript presented in an intelligible fashion and written in standard English?

Reviewer #1: Yes

Reviewer #2: Yes

Reviewer #1: (No Response)

Reviewer #2: All revisions have been addressed completely and thoroughly. Thank you to the author(s) for their efforts in preparing this revision.

**Do you want your identity to be public for this peer review?** For information about this choice, including consent withdrawal, please see our Privacy Policy

Reviewer #1: **Yes:** Valerie Johnson

Reviewer #2: No

---

## [Editor Report · Acceptance letter]

PONE-D-25-54442R1

PLOS One

Dear Dr. Stacy,

I'm pleased to inform you that your manuscript has been deemed suitable for publication in PLOS One. Congratulations! Your manuscript is now being handed over to our production team.

Kind regards,

on behalf of

Dr. Vitor Hugo Rodrigues Paiva

Academic Editor

PLOS One